

# Plant compensation to grazing and soil carbon dynamics in a tropical grassland

Mark E. Ritchie

Department of Biology, Syracuse University, Syracuse, NY, USA

## ABSTRACT

The effects of grazing on soil organic carbon (SOC) dynamics, particularly in the tropics, are still poorly understood. Plant compensation to grazing, whereby plants maintain leaf area (C input capacity) despite consumption (C removal) by grazers, has been demonstrated in tropical grasslands but its influence on SOC is largely unexplored. Here, the effect of grazing on plant leaf area index (LAI) was measured in a field experiment in Serengeti National Park, Tanzania. LAI changed little for grazing intensities up to 70%. The response curve of LAI versus grazing intensity was used in a mass balance model, called SNAP, of SOC dynamics based on previous data from the Serengeti. The model predicted SOC to increase at intermediate grazing intensity, but then to decline rapidly at the highest grazing intensities. The SNAP model predictions were compared with observed SOC stocks in the 24 grazed plots of a 10-year grazing exclosure experiment at eight sites across the park that varied in mean annual rainfall, soil texture, grazing intensity and plant lignin and cellulose. The model predicted current SOC stocks very well ($R^2 > 0.75$), and suggests that compensatory plant responses to grazing are an important means of how herbivores might maintain or increase SOC in tropical grasslands.

Subjects Ecology, Ecosystem Science, Environmental Sciences, Soil Science
Keywords Herbivory, Grazing, Plants, Compensation, Leaf area, Soil carbon, Modeling, Grasslands, Tropical, Serengeti

## INTRODUCTION

Soil organic carbon (SOC) in grasslands and savannas represents one of the largest reservoirs of carbon on earth (*Conant et al., 2001*; *Lal et al., 2007*). Due to widespread unsustainable land use due to overgrazing, loss of native herbivores, excessive fires, and the potential to reverse the impacts of these uses, SOC is one of the more important potential sinks of greenhouse gases in the effort to mitigate climate change (*Mannetje, 2007*; *Ellis & Ramankutty, 2008*; *Smith et al., 2008*). A major question is what management practices in grasslands, in the form of grazing, fire, fertilization, re-vegetation and restoration, etc., can lead to net sequestration of carbon.

Recent reviews demonstrate that herbivores can have dramatically different effects on soil organic carbon (SOC), both positive and negative, depending on soil type, precipitation, plant species composition and grazing intensity (*Milchunas & Lauenroth, 1993*; *Derner & Schuman, 2007*; *Pineiro et al., 2010*; *McSherry & Ritchie, 2013*). These results have been obtained largely in temperate grasslands grazed by livestock (*McSherry & Ritchie, 2013*), and relatively little is known about grazing impacts on SOC in tropical

Corresponding author
Mark E. Ritchie, meritchi@syr.edu

grasslands. In the tropics and in certain temperate climates, plant species composition is dominated by warm-season (C4) grasses that, among other adaptations to grazing, invest heavily in rhizomes and other storage organs that allow them to respond quickly to both rainfall and defoliation (*McNaughton, 1985*; *Milchunas & Lauenroth, 1993*; *Dubeux et al., 2007*). Compensatory responses to grazing can involve sacrificing stems for leaves and thus maintain leaf area and potential carbon inputs despite carbon off-take by herbivores (*Anderson, Dong & McNaughton, 2006*; *Zheng et al., 2010*; *Ziter & MacDougall, 2013*). Secondly, C4 grasses generally contain higher levels of lignin and cellulose (*Barton et al., 1976*), which are generally recalcitrant to decomposition, and produce extensive aboveground litter that frequently burns during extended dry seasons (*Trollope, 1982*; *McNaughton, Stronach & Georgiadis, 1998*; *Sankaran, Ratnam & Hanan, 2008*). Thirdly, benign temperatures allow for the prevalence of macro-decomposers, such as termites and dung beetles, to rapidly incorporate senesced plant material and herbivore dung into soil (*Hanski & Cambefort, 1991*; *Freymann et al., 2008*; *Dungait et al., 2009*; *Risch, Anderson & Schutz, 2012*). These features may dramatically alter herbivore impacts on SOC in tropical grasslands as compared to temperate grasslands with equivalent precipitation and soil type (*McSherry & Ritchie, 2013*).

Of these three important features, the consequence of plant compensation for soil carbon is probably the least well understood. Leaf area index (leaf area/area sampled), or LAI, has been shown to decrease in response to grazing in some contexts but not others (*Anderson, Dong & McNaughton, 2006*; *Zhao et al., 2009*; *Cruz et al., 2010*; *Zheng et al., 2010*; *Ziter & MacDougall, 2013*). Studies in Serengeti National Park (*McNaughton, 1985*; *McNaughton, Milchunas & Frank, 1996*) and west Africa (*de Mazancourt, Loreau & Abbadie, 1999*) have shown biomass compensation to grazing, but not compensation for lost leaf area specifically. Furthermore, the quantitative manner in which LAI changes as a response to varying grazing intensity has not been fully explored. A major question is whether LAI can be maintained or even increase with increasing grazing intensity and at what level of grazing intensity it declines.

The response of LAI to herbivory may be a key driver of soil carbon response to grazing. Leaf area is highly correlated with gross photosynthesis and net carbon (C) assimilation (*Craine et al., 2001*; *Ellsworth et al., 2004*; *Reich, Wright & Lusk, 2007*). Maintenance of leaf area following defoliation potentially allows C inputs into important carbon pools, such as biomass and SOC, to be little affected by herbivory, while loss of leaf area can dramatically reduce C inputs. Because most current soil carbon dynamic models lack an explicit accounting of herbivore impacts on leaf area (*Paustian, Parton & Persson, 1992*; *Schimel et al., 1994*; *Parton et al., 1995*), the relative importance of leaf area-related C inputs relative to C losses from fire, decomposition, and herbivore consumption have not been explored.

The goal of this paper is to show how leaf area index (LAI), or leaf area/area sampled, and plant allocation to stem and leaf changes with grazing intensity in a tropical savanna and then to explore how such possible compensatory responses might affect soil carbon dynamics. Hence, this paper (1) presents the results of a 2-year grazing exclosure

experiment in Serengeti National Park to measure effects of varying grazing intensity on plant LAI and proportion of biomass allocated to stems vs. leaves, (2) develops a model of soil carbon dynamics based on extensive past research in the Serengeti as a framework for evaluating the impact of LAI response to grazing, and (3) compares predictions of this model with observed soil carbon stocks across the eight sites of a 9-year grazing exclosure experiment in which key model input parameters were independently measured.

The model of carbon dynamics, called SNAP (referring to its origin in Serengeti National Park) accounts explicitly for fixed carbon, as determined by LAI, soil texture and rainfall, and its allocation to roots versus shoots, and its fate: decomposition, consumption by herbivores, combustion in fire or residence in a long-term recalcitrant SOC pool. The model has five input variables: mean annual rainfall, grazing intensity, fire frequency, aboveground proportion of cellulose + lignin, and soil texture (percent sand), based on the suggested most important fates of stable carbon in the Serengeti ecosystem (*McNaughton, 1985*; *McNaughton, Banyikwa & McNaughton, 1998*; *McNaughton, Stronach & Georgiadis, 1998*; *Holdo et al., 2007*, *Holdo et al., 2009*). The model and its functional relationships are populated with parameter values available from a substantial literature on grazing, plants, soils, and microbes from Serengeti National Park plus some additional new data on grazing intensity, plant lignin and cellulose and soil moisture gathered from a long-term (9 year) grazing experiment across eight sites in Serengeti National Park that vary in grazing intensity, rainfall, soil texture, fire history, and plant species composition.

To evaluate the role of plant compensation, a sensitivity analysis of model parameters, including grazing intensity, in the SNAP model is conducted. The model also is used to make deductive predictions of current soil carbon stocks under the assumption of a mass balance steady-state resulting from the persistence of mean rainfall, plant lignin and cellulose, soil texture, fire history, and grazing intensity conditions over an extended period of prior years. These predictions are then compared with observed soil carbon measured in grazed plots of the 9-year exclosure experiment using linear regression, with tests for overall model fit ($R^2$) and tests for differences in slope from a value of 1 and an intercept value of zero. This analysis provides an explicit and quantitative test of the hypothesis that plant compensatory responses are important in the influence of herbivores on soil carbon.

This approach applies a deductive (*Belovsky, 1984*; *Belovsky, 1994*; *Overmars, de Groot & Huigen, 2007*) rather than inductive (*Burnham & Anderson, 2004*) evaluation of the impact of possible plant compensation. The use of a deductive model explicitly combines the logically-derived hypothesized mechanisms by which different factors might drive soil carbon dynamics based on prior knowledge of the Serengeti. A sensitivity analysis of the model assesses the relative importance of a given variable or parameter relative to others. The advantage of a deductive approach is that it avoids the uncertainty in inference derived from associating soil carbon dynamics with multiple, possibly auto-correlated, independent variables in an inductive approach (*Burnham & Anderson, 2004*). Further, a deductive model builds in non-linearity of relationships, which is largely avoided in inductive approaches but may be likely to occur in the relationship between

LAI and grazing (*McNaughton, 1985*; *de Mazancourt, Loreau & Abbadie, 1999*; *Anderson, Dong & McNaughton, 2006*).

## MATERIALS AND METHODS

### Study area

Most of the data presented in this study were collected in Serengeti National Park (SNP), Tanzania. SNP contains nearly 3 million migratory wildebeest (*Connochaetes taurinus*), zebra (*Equus burchelli*), and Thomson's gazelles (*Gazella thomsoni*), plus multiple species of resident herbivores >10 kg in size, all distributed heterogeneously (*Anderson et al., 2010*) over 25,000 km$^2$ (a mean density of 120/km$^2$) (*McNaughton, 1985*; *Sinclair et al., 2007*). These herbivores consume a major fraction (mean 61.8% $\pm$ 12.2 SEM) of aboveground primary production (*McNaughton, 1985*). Vegetation is dominated by C$_4$ grasses, featuring *Themeda triandra* in more lightly grazed areas and *Digitaria macroblephara* and *Pennisetum menzianum* in more heavily grazed areas. The Serengeti region features a pronounced rainfall gradient driven mixed somewhat independently with a soil gradient of silty, organic matter-rich soils in the southeast to clay soils in the west and sandy, low nutrient soils in the north (*Anderson et al., 2007*; *Sinclair et al., 2007*; *Anderson et al., 2010*). The clustered distribution of resident herbivores (*Anderson et al., 2010*) and variable movements of migratory species results in highly variable historical grazing intensity across the landscape. Strong preferences by herbivores for plants found at the top rather than bottom of slopes plus a tendency to avoid the edges of woodlands results in variable grazing intensity over distances of <50 m (*Gwynne & Bell, 1968*; *Anderson, Dong & McNaughton, 2006*; *Anderson et al., 2007*).

### Design

Plant compensation to grazing was assessed near the Serengeti Wildlife Research Centre near Seronera (34° 50′ E and 2° 25′ S) inside the Park. The area is open *Acacia tortilis* savanna woodland, with dominant grasses *Themeda triandra* and *Digitaria macroblephara*, and hosts resident herds of impala (*Aepyceros melampus*), buffalo (*Syncerus caffer*), and Thomson's gazelles (*Gazella thomsoni*), among others, and is routinely visited by the annual wildebeest and zebra migration. Plant responses to grazing were measured inside and outside of grazing exclosures established 2 years previously in a 10 ha open grassland with less than 5% tree cover near the Serengeti Wildlife Research Centre (2-year exclosure experiment). The proportion of biomass as stem versus leaf was measured for all plants clipped from six randomly selected 15 $\times$ 15 cm quadrats, three inside and three outside each of twelve exclosures of various size (10–200 m$^2$) erected between 2009 and 2010 to protect experimental gardens or nutrient addition experiments. LAI, proportion of stem, and aboveground biomass were measured in the unused "buffer zone" just inside each fence and then in quadrats 4 m perpendicular to each fence paired with each quadrat inside the fence. The objective was to encounter different localized grazing intensities that resulted from sampling beneath thorny shrubs or other plant protections versus patches with highly preferred forage species.

To generate additional input data for a model of soil carbon dynamics, a longer-term (9 year) grazing exclosure experiment established by Sam McNaughton in 1999 (*Anderson, Ritchie & McNaughton, 2007*) was used to provide data on soil moisture as a function of rainfall, and independent measures of grazing intensity, plant lignin and cellulose, and SOC to test the SOC dynamic model. A line of six plots (4 × 4 m), spaced 10 m apart in a line transect were established in October 1999 at each of 8 sites located 10 or more km apart, in grassland areas visited almost entirely by grazing, rather than browsing, ungulate species (*Anderson, Ritchie & McNaughton, 2007*). Sites were chosen to be within 1 km of large, permanent concentrations of grazing herbivores (*Anderson et al., 2010*) and varied considerably in annual rainfall, soil type, and fire frequency. Three randomly selected plots within the transect at each site were fenced with 2 m high, 8 cm mesh wire. These fences effectively excluded all grazing mammals >10 kg, since animals preferred to go around rather than jump fences. The remaining three plots at each site were unfenced controls. Woody vegetation at our experimental sites was typically sparse (less than 10 woody plants/ha) but herbaceous legumes, such as *Indigofera volkensii*, were ubiquitous and reasonably abundant (>2% cover) at all our study sites. Our eight sites included two on the treeless shortgrass plains of southeastern SNP, and the other six sites were representative of woody savannas broadly distributed across SNP (*Ruess & Seagle, 1994*).

### Measurements

The effect of grazing on plant (grass) allocation to leaf versus stem and on leaf area index, LAI ($cm^2$ leaf/$cm^2$ ground area) was assessed. Clipped plants for all plant species in a quadrat were taken back to the lab and, while fresh, laid out and traced on graph paper, and then leaves were clipped from stems and stored, dried at 45°C, and weighed separately (as with field biomass). The proportion of leaf for each quadrat was calculated, along with LAI (total leaf area ($cm^2$) divided by quadrat area ($cm^2$)).

Another key variable, GI (grazing intensity) varies locally and was estimated in both the 2-year and 9-year experiments as the fractional difference in standing aboveground biomass (AGB) : $GI = 1 - AGB_g/ABG_{ug}$ (*Anderson, Ritchie & McNaughton, 2007*). GI is an indirect measure of the fraction of annual production consumed by grazers (*McNaughton, 1985*; *McNaughton, Milchunas & Frank, 1996*). AGB was measured by clipping and weighing all aboveground plant material, excluding litter, after drying at 45°C for three days to constant mass. AGB was measured in the 2-year exclosure experiment in the paired quadrats inside and outside exclosures used to measure LAI. The quadrat inside the exclosure was assigned GI = 0, and the paired quadrat outside the exclosure was assigned GI based on the difference in AGB between the paired quadrats. In the 9-year exclosure experiment, biomass was clipped from four 15 × 15 cm quadrats within each grazed and exclosure plot in late May or June at the end of the wet season in 2006, 2009, and 2010. Biomass measured in each of the three grazed plots at each site was compared to biomass in each of their nearest neighbor exclosure plots (*Anderson, Ritchie & McNaughton, 2007*), yielding three estimates of grazing intensity at each site.

To measure lignin pluscellulose fraction (LIGCELL), dried plant material from the entire clipped quadrat was ground through a 0.9 mm (40 mesh) screen in a Wiley mill

and then subjected to a sulfuric acid hydrolysis method (*Sluiter et al., 2010*), i.e., sequential digestion in first neutral detergent manufactured by Ankom Technology Corp., Macedon, New York, USA (sodium lauryl sulfate, 50.02%, EDTA disodium, dehydrate, 31.03%., sodium borate, decahydrate 11.35%, and sodium phosphate, dibasic, anhydrous, 7.6%), followed by Ankom manufactured acid detergent (20 g cetyl trimethylammonium bromide (CTAB) diluted in 1 L 1.00 $NH_2SO_4$), and then in 72% sulfuric acid, followed finally by ashing in a muffle furnace at 500°C for 24 h. Lignin and cellulose fractions were inferred by the mass lost between the remainder following neutral detergent digestion and the remainder after ashing, expressed as a proportion of the original sample mass.

Annual rainfall over the 9 years of the study was obtained from rain gauge data at 23 sites collected by the Serengeti Ecological Monitoring Program (*Anderson et al., 2007*). Fire frequency was determined by interpreting MODIS satellite visible and near infra-red imagery (*Dempewolf et al., 2007*) for July–October for 2000–2008 to generate maps of fire occurrence for each year with a resolution of 250 m. Grazing experiment site GPS locations were overlaid with fire maps for each year in a GIS (ArcGIS 9.1 by ESRI®) to determine the number of times a site burned over the period 2000–2008.

Effects of precipitation on soil moisture and number of days in which microbes were likely to be active (WETDAYS) were determined from field measurements of soil moisture in the long-term experiment (unpublished data generously provided by Sam McNaughton). Five cm diameter soil cores were taken to a depth of 10 cm each month from December 1999 to June 2002 (a total of 31 days) at each of three grazed plots at the eight sites of the long-term Serengeti grazing experiment. Moisture content was determined gravimetrically and monthly rainfall was measured from rain gauges mounted on 1 m tall fence posts at each site. A recent review (*Manzoni, Schimel & Porporato, 2012*) suggests that, in semi-arid soils, microbial activity ceases when water potential drops below −14 cm $H_2O$. Water potential was converted into soil moisture, calibrated for the texture of Serengeti soils (Table 1) (*Cosby et al., 1984*) and suggested that *Manzoni, Schimel & Porporato (2012)* threshold corresponded on average to 10% gravimetric water content. The number of days at each of the eight sites that soil moisture was 10% or higher was determined and then used to build a regression to estimate the number of days per year with active soil microbes as a function of annual rainfall.

As a test of the new soil carbon model, SOC was measured at the same time as biomass in each plot from 8.3 cm diameter × 40 cm deep cores from each of the three biomass quadrats per plot. Each core was spread on a portable table top and crushed with a rolling pin to break up soil aggregates. All pebbles, rocks, and visible roots were removed, and then soil was sieved through 0.7 mm mesh. Cores of known volume were pooled, stored in sealed plastic bags, and then dried for 6 days at 45°C and then weighed to determine bulk density (g soil/cm$^3$). Both SOC and nitrogen contents were determined with the Walkley-Black method at the Soils Analysis Laboratory at Sokoine University of Agriculture in Morogoro, Tanzania. Reported error was ±0.004 g/gC. Soil composition (sand, silt and clay fractions) was determined by a micropipette method. Soil carbon density was measured by multiplying SOC content by bulk density (g/m$^2$) to 40 cm depth

**Table 1 Variables and key parameters in the SNAP model, their units, and sources of data.**

| Parameter | Units | Type | Source |
|---|---|---|---|
| $ANPP^{max}$ | $g\,m^{-2}\,yr^{-1}$ | Literature | Maximum aboveground production (*McNaughton, 1985*) |
| $ANPP^{est}$ | $g\,m^{-2}\,yr^{-1}$ | Model | Grazer-modified ANPP |
| $BNPP^{est}$ | $g\,m^{-2}\,yr^{-1}$ | Literature | Belowground production (*McNaughton, Banyikwa & McNaughton, 1998*) (Fig. 3B) |
| DDSOC | $gC\,m^{-2}\,yr^{-1}$ | Model | Dung-derived SOC |
| FIRE | #/yr | Measured | Number of fires per year, monitored (Table 4) |
| GI | proportion | Measured | 1- (grazed biomass/ungrazed biomass) (Table 4) |
| LAI | - | Measured | Leaf Area Index (dimensionless); This study (Fig. 2B) |
| LIGCELL | proportion | Measured | Cellulose + lignin (%) aboveground biomass, (Table 4) |
| MRESP | $gC\,m^{-2}\,d^{-1}$ | Literature | (*Ruess & Seagle, 1994*) (Fig. 3D) |
| $P_L$ | proportion | Measured | Proportion leaf; This study (Fig. 2A) |
| PDSOC | $gC\,m^{-2}\,yr^{-1}$ | Model | Plant-derived SOC |
| RAIN | mm/yr | Interpolated | Mean annual rainfall (*Anderson et al., 2010*) (Table 4) |
| SAND% | percent | Measured | From a standard soil analysis, measured (Table 4) |
| WETDAYS | $d\,yr^{-1}$ | Measured | #days with soil >10% water; This study (Fig. 3C) |
| WHC | proportion | Literature | Water holding capacity (*Ruess & Seagle, 1994*) (Fig. 3A) |

**Table 2 Parameters for Monte Carlo simulations.** Parameters and their errors used in predicting soil carbon stocks in Serengeti grasslands and in the Monte Carlo simulations.

| Parameter | Input | Coefficient | | | | | |
|---|---|---|---|---|---|---|---|
| | | $B_0$ | SE | $B_1$ | SE | $B_2$ | SE |
| $P_L$ | GI | 0.597 | 0.061 | 0.24 | 0.023 | | |
| LAI | GI | 1.15 | 0.027 | −0.015 | 0.0011 | 4.6 | 0.2 |
| $ANPP^{max}$ | RAIN | −27.5 | 17.6 | 0.84 | 0.07 | | |
| WHC | SAND% | 1.04 | 0.05 | −0.0070 | 0.0008 | | |
| $BNPP^{est}$ | RAIN | 958.8 | 165.0 | −0.82 | 0.27 | | |
| WETDAYS | RAIN | −0.025 | 0.033 | 0.00043 | 0.00006 | | |
| MRESP | SOC | −0.58 | 0.45 | 0.00044 | 0.00007 | | |
| CARBON | N/A | 0.45 | 0.02 | | | | |

of the pooled cores in each plot. A 40 cm depth was used, as this typically corresponds to the depth of >90% of roots (*McNaughton, Stronach & Georgiadis, 1998*), and multiple sites featured a hardpan or dense clay layer at 40–50 cm that prevented reasonable sampling to greater depths.

## Literature data and re-analyses

Literature data, mostly from prior studies in Serengeti, were re-analyzed to specifically estimate soil water holding capacity (WHC), aboveground production, belowground production, and soil microbial respiration of carbon. For sources and units of literature data, see Table 1. The relationships used are reported in the Results and Discussion (Table 2).

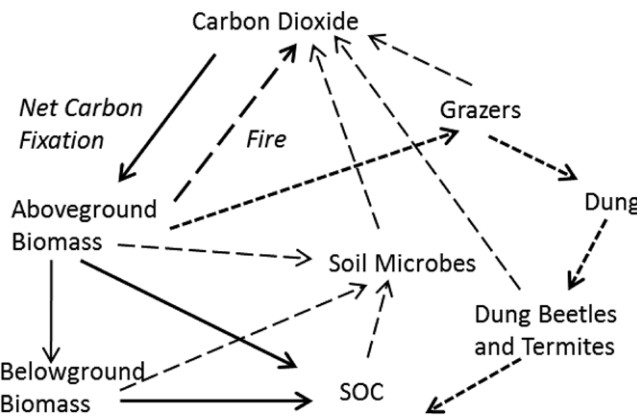

**Figure 1 Hypothetical major fates of carbon in tropical grassland, as the basis for a practical soil carbon dynamic model.** Net fixed carbon becomes resident soil organic carbon (SOC) through two major pathways: plant-derived SOC not consumed by microbes (in soil or the guts of macro-decomposers like termites) (heavy solid arrows), and dung-derived SOC (heavy short-dashed arrows) not assimilated by grazers and gut or soil microbes. All other carbon is combusted through fire (black arrow), or respired by grazers and microbes in soil or the guts of macro-decomposers (long dashed arrows).

## Model development

The flow of fixed carbon through hypothetical tropical grassland was described according to four criteria. First, mass balance was assumed, so that all fixed carbon resided in pools, such as biomass or SOC, or suffered the fate of being combusted, consumed, or respired. Second, this flow was described in the minimum number of fluxes and pools necessary to account for the fate of carbon (Fig. 1). Third, the most parsimonious set of input variables that could describe the changes in these pools was used. Finally, a time step of one year with a distinct wet and dry season each year was assumed, with production and grazing resulting in a standing biomass at the beginning of the dry season each year that subsequently either combusted if there was fire or was otherwise decomposed. More complicated time dynamics of daily or seasonal changes have been used recently (*Holdo et al., 2007*, *2009*), but this introduces possibly unnecessary complexity. Above- and belowground production are therefore seasonal accumulations of biomass as modified by average grazing intensity over the course of the wet season. Biomass not consumed by herbivores may then burn at some frequency during the dry season. These assumptions led to a relatively simple set of pathways for stable carbon (Fig. 1): (1) input into soil organic matter from decomposing shoots or roots, (2) combustion in fire, or (3) input into soil organic matter through dung. All non-lignin or cellulose compounds were assumed to turnover within 1–2 years or less (*Ruess & Seagle, 1994*) and were consequently ignored.

The model was based on four major assumptions. First, tropical grasses potentially exhibit compensatory responses to defoliation that maintain similar leaf area, and thus photosynthetic capacity, across a broad range of grazing intensities (*McNaughton, 1985*). Second, the largest carbon inputs to the soil organic matter pool occur through

decomposition of above- and belowground biomass and through incorporation of herbivore dung into soil. Third, the major losses of SOC derive from combustion (fire), herbivore respiration and soil microbial respiration. Finally, all plant C in plant tissue other than lignin and cellulose was assumed to be assimilated and respired by herbivores, microbes in soil or the guts of macro-decomposers. Lignin and cellulose are the most recalcitrant forms of carbon to decomposition, and likely account for the majority (>90%) of long-lived SOC (*Ganjegunte et al., 2005*; *Frank, Pontes & McFarlane, 2012*).

The SOC (g/m$^2$ to 40 cm depth) measured in 2009 in the grazed plots in the long-term grazing experiment represented SOC$_{eq}$ under the reasonable assumption that conditions in these plots had been approximately similar in terms of wildlife use, plant species composition, and climate since 1979 (*Sinclair et al., 2007*; *Holdo et al., 2009*). The average of grazing intensities measured in 2006, 2009, and 2010 was assumed to reflect historical long-term grazing. The other input variables either were already a long-term average (RAIN, FIRE) or are unchanged by land use (SAND%).

### Statistics

The association between LAI and grazing intensity (GI) was tested with linear regression in SPSS version 19 (IBM, Armonk, New York, USA). Non-linear relationships were also explored, and based on the appearance of the data, the R script "nls" was used with a function $B_0 - B_1 \exp(-B_2 \text{ GI})$, where $B_i$ are fitted coefficients. Linear regression in SPSS was used to build regression relationships from literature data (Table 2) and to compare observed SOC to predicted SOC$_{eq}$ from the SNAP model. In the comparison of the observed and predicted SOC, the hypotheses that the intercept of the regression line was equal to zero and that the slope was equal to one were evaluated with z-tests.

The SNAP soil carbon dynamic model was constructed from a number of empirical relationships that contain parameters (intercepts, slopes, etc.) estimated with error. A Monte Carlo analysis (*Ogle et al., 2010*) was conducted by sampling the normal distributions inferred from measured model input parameters and parameters from relationships used to construct the model. This process was repeated 100 times to generate a mean and s.e.m. for the estimated SOC$_{eq}$ for each site and each paired grazed and ungrazed plot in the 9-year exclosure experiment.

## RESULTS AND DISCUSSION

### New and prior data analyses

The proportion of biomass in leaves increased significantly ($P < 0.001$) with increasing grazing intensity (Fig. 2A). In concordance with this pattern, Leaf Area Index (LAI) declined significantly with increasing grazing intensity (GI) according to linear regression ($R^2 = 0.36$, df = 1,34, P = 0.01). However a non-linear function, LAI = $B_0 - B_1 \exp (B_2 \text{ GI})$ fit the data much better ($R^2 = 0.58$, df = 1,33, $P < 0.001$) and visibly matched better the consistently high LAI found at low to intermediate grazing intensity and the dramatic decline in LAI above a grazing intensity of 70% (Fig. 2B). These two patterns support the hypothesis that plants maintain C inputs to the Serengeti ecosystem despite

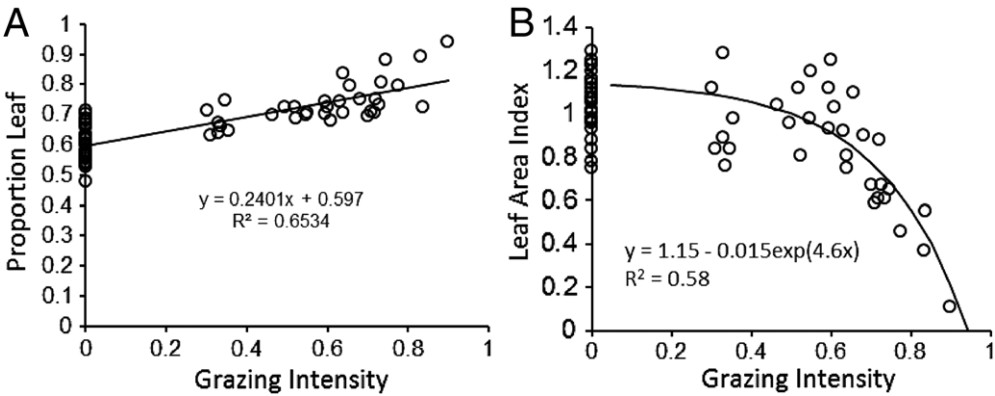

**Figure 2 Plant compensatory responses to grazing in Serengeti.** Key relationships for assessing plant compensation to grazing: (A) proportion of leaves ($P_L$) and (B) Leaf Area Index (LAI, $cm^2/cm^2$) of grazed and ungrazed plants, each as a function of grazing intensity (GI).

heavy removals of biomass by grazers. The maintenance of LAI across such a broad range of grazing intensities contrasts with the typical pattern of dramatic reductions in LAI seen mostly in temperate systems under relatively moderate grazing intensities (30%–60%) (*Zhao et al., 2009*; *Cruz et al., 2010*; *Zheng et al., 2010*). The pattern corroborates results of enhanced productivity under grazing (*McNaughton, 1985*; *Milchunas & Lauenroth, 1993*; *Frank, Kuns & Guido, 2002*) that are at least partly derived from compensatory morphological changes by plants to grazing (*Anderson, Dong & McNaughton, 2006*; *Ziter & MacDougall, 2013*). Despite these previous results, it is still not clear why the $C_4$ grasses typically dominant in the Serengeti and elsewhere respond differently to moderately intense grazing than $C_3$ grasses (*Derner & Schuman, 2007*; *McSherry & Ritchie, 2013*).

As expected, water holding capacity exhibited a strong, significant ($P < 0.001$) negative association with increasing sand content (*Ruess & Seagle, 1994*) (Fig. 3A), and root production showed a significant ($P = 0.0015$) decline in association with higher rainfall (*McNaughton, Banyikwa & McNaughton, 1998*) (Fig. 3B). The proportion of days soil held more than 10% moisture increased significantly ($P = 0.006$) with mean annual rainfall across the eight long-term grazing experimental sites (Fig. 3C). Finally, maximum microbial respiration rates were strongly and significantly ($P < 0.001$) associated with existing SOC stocks across the Serengeti landscape (*Ruess & Seagle, 1994*) (Fig. 3D). Regression equations are reported in the figures and were used in the development of the SOC dynamic model.

## SOC dynamic model

Based on the results for LAI response to grazing (Fig. 2B), the SOC model for Serengeti National Park (hereafter referred to as the SNAP model) was constructed with five input variables: RAIN, mean annual rainfall (mm/yr); GI, grazing intensity (1-(grazed biomass/ungrazed biomass)) (*McNaughton, 1985*); FIRE, fire frequency or the number of fires recorded, divided by the number of years over which fires were monitored;

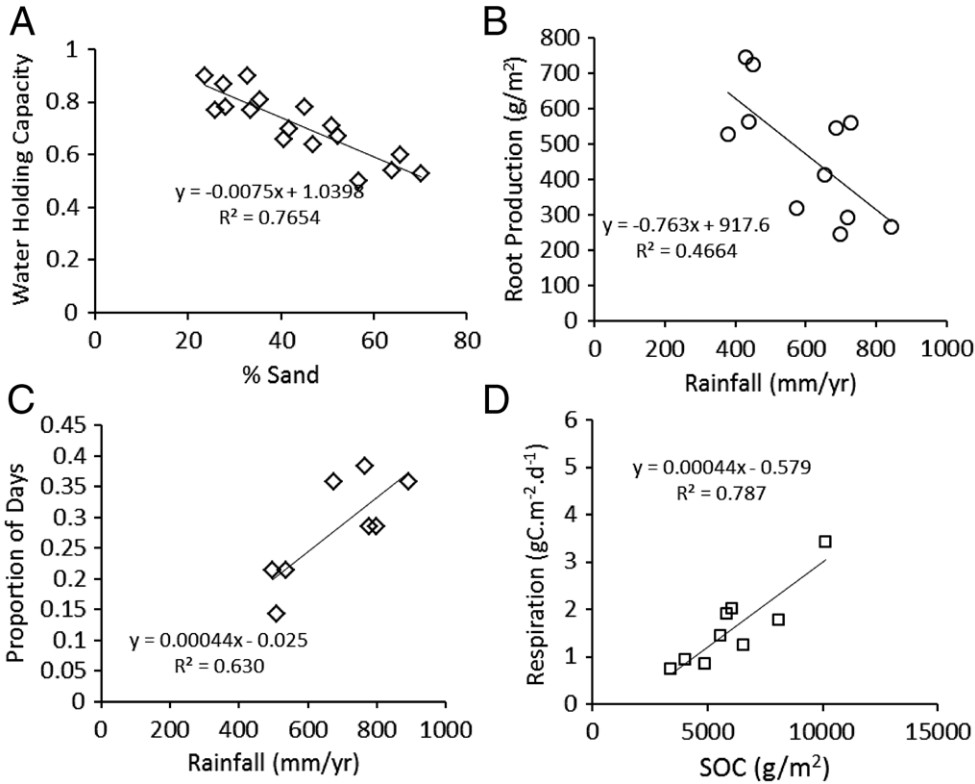

**Figure 3 Functional relationships used to build the SNAP soil carbon model.** Key analyses of literature data to build empirical relationships for key model parameters. (A) the decline in water-holding capacity of soils with increasing sand content (*Ruess & Seagle, 1994*), (B) annual belowground production as a function of rainfall (*McNaughton, Stronach & Georgiadis, 1998*), (C) number of moist soil days (WETDAYS) as a function of annual rainfall (RAIN), and maximum microbial maximum respiration rates as a function soil carbon stocks (SOC) (from *Ruess & Seagle (1994)*).

LIGCELL, lignin and cellulose content (proportion) in aboveground herbaceous vegetation, and SAND%, (*McNaughton, 1985*). The sources of these data are summarized in Table 1 and specific regression equations with s.e.m. error terms are provided in Table 2.

Two additional factors that can be important influences on carbon dynamics, nitrogen (*Pineiro et al., 2010*) and soil temperature (*Miller et al., 2004*), were also considered but not included as inputs into the SNAP model. Production in the Serengeti is assumed to be co-limited by nitrogen and water, and total soil $N$ was included in a validated model of plant productivity for Serengeti (*Holdo et al., 2007*). Simplifying the functional forms used in Holdo et al. s model and solving at steady state revealed that ANPP$^{max}$ is likely to be proportional to the product of soil N and rainfall. However a multiple regression of the logarithms of ungrazed biomass (production) in the long-term grazing exclosures against the logarithms of soil N (ln(SoilN) and mean annual rainfall (ln(RAIN)) revealed that ln(RAIN) was a highly significant factor, but ln(Soil N) was not ($P = 0.45$). This outcome
suggests that N is a far less important factor influencing productivity in the Serengeti than is rainfall. In addition, soil N or soil C:N was not a significant variable explaining residual variation in microbial respiration in Serengeti soils following regression if SOC was included in a regression model (partial $R^2 = 0.05$, N $= 16$, $P < 0.41$). Finally, the influence of other variables that are often correlated with higher N, such as finer-textured soils (*Ruess & Seagle, 1994*) and more intense grazing (*Holdo et al., 2007*) will already be included in the model in other relationships.

The influence of soil temperature on microbial respiration was also not incorporated into the model. Temperature in most tropical soils is not as variable as in temperate systems, typically ranging over only 3°C over an entire year (*Caquet et al., 2012*), and increased soil temperature is typically accompanied by drier conditions, thus canceling any temperature-related increases. Consequently, temperature was viewed as a difficult-to-measure parameter that would likely add relatively little explanatory power to the model.

Fixed carbon was accounted for as biomass produced during the wet season, and was assumed to be allocated either aboveground or belowground. Estimated aboveground production in year $t$, $ANPP_t^{est}$, was determined as potential production, modified by grazing impacts on proportion biomass as leaf and specific leaf area. Potential production as a function of rainfall (Table 2) was derived from *McNaughton (1985)* empirical relationship and weighted by the adjusted water holding capacity (AWHC), as determined by soil sand content AWHC $= 1.33 - 0.0075*$SAND%. This is the relationship between water holding capacity (WHC) and SAND% derived from *Ruess & Seagle (1994)* $(1.08 - 0.0075*$SAND%) but with 2 standard deviations in their measured WHC added to the intercept. This converted WHC into a coefficient that adjusted *McNaughton (1985)* productivity relationships, which reflected average soil properties, for whether soil is higher or lower than average in sand content. Adjusting potential productivity for water holding capacity incorporates how soil influences soil moisture for plant growth at a given rainfall amount.

Thus:

$$ANPP_t^{max} = (0.84 * RAIN - 27.5) * (1.33 - 0.0075 * SAND\%). \quad (1)$$

Potential aboveground production is then adjusted by leaf area index, LAI, to calculate estimated productivity $ANPP_t^{est}$, which accounts for the stimulation of leaf area at the expense of loss of stem due to grazing.

$$ANPP_t^{est} = LAI * ANPP_t^{max}. \quad (2)$$

As measured by *McNaughton (1985)*, $ANPP_t^{est}$ can be higher than $ANPP_t^{max}$ at low to intermediate grazing intensities because grazing keeps plants at a faster-growing stage during the wet season. At high grazing intensities, leaf area declines and $ANPP_t^{est}$ approaches zero. LAI was determined from the non-linear empirical relationship for LAI measured in this study (Fig. 2B):

$$LAI = P_L/0.6 - 0.015 * EXP(4.6 * GI), \quad (3)$$

where $P_L$ is the proportion of biomass as leaves, GI is grazing intensity, and EXP is the exponential function. $P_L$ was also estimated from plant compensation to grazing measurements (Fig. 2A)

$$P_L = 0.6 + 0.24 * GI. \tag{4}$$

The ratio $P_L/0.6$ in Eq. (3) incorporates the grazing-induced adjustment in the proportion of leaves into the calculation of LAI, such that LAI $= 1$ at GI $= 0$ and $P_L = 0.6$, the y-intercept value in Fig. 2A. Combining Eq. (2)–(4) yields estimated aboveground productivity as a function of rainfall, soil texture, and grazing intensity:

$$\begin{aligned} ANPP_t^{est} &= \{[(0.84 * RAIN - 27.5) * (1.33 - 0.0075 * SAND\%)] \\ &\quad * [(0.6 + 0.24 * GI)/0.6 - 0.015 * EXP(4.6 * GI)]. \end{aligned} \tag{5}$$

Belowground production in year $t$ generally declines with increasing rainfall (*McNaughton, Banyikwa & McNaughton, 1998*) (Fig. 1C)

$$BNPP_t^{est} = 917.4 - 0.763 * RAIN. \tag{6}$$

With these C input functions, inputs to SOC were calculated from the two major sources, *plant-derived carbon* $PDSOC_t$ (solid arrows in Fig. 1) and *dung-derived carbon*, $DDSOC_t$ (thick short-dashed arrows in Fig. 1). Plant-derived carbon consists of the carbon fraction of lignin and cellulose in roots plus the carbon fraction of lignin and cellulose in aboveground biomass that is not consumed by grazers or burned. Total non-ash-free carbon content of aboveground plant tissue (*Tao et al., 2012*) averages 0.45. Lignin and cellulose in roots was not measured but was estimated to be 5% higher than that in aboveground tissues (*Andrioli & Distel, 2008*; *Semmartin, Garibaldi & Chaneton, 2008*). Grazing and fire interact in that fire can only burn biomass not consumed by grazers, so that aboveground decomposition is only of the fraction of biomass not grazed (1-GI) and not burned (1-FIRE). Fire and grazing were assumed to have no important effects on belowground production (*McNaughton, Banyikwa & McNaughton, 1998*).

$$\begin{aligned} PDSOC_t &= 0.45 * [LIGCELL * ANPP_t^{est} * (1 - GI) * (1 - FIRE) \\ &\quad + (LIGCELL + 0.05) * BNPP_t^{est}]. \end{aligned} \tag{7}$$

Dung-derived carbon was assumed to be the lignin and cellulose fraction of biomass consumed by grazers. All stable C in dung was assumed to be incorporated in soil because of the activities of termites and dung beetles, which often incorporate dung into soil within 2–5 days (*Freymann et al., 2008*; *Risch, Anderson & Schutz, 2012*), allowing little C to volatilize. While a significant fraction of the dung is consumed and respired by termites and dung beetles, the stable C fraction likely will remain intact as soil organic matter and thus contribute to SOC. As for aboveground plant tissue, C content of dung was 0.45.

$$DDSOC_t = LIGCELL * 0.45 * GI * ANPP_t^{est}. \tag{8}$$

Carbon losses associated with microbial respiration ($MRESP_t$) were accounted for as a function of the number of days in which soil moisture exceeds a threshold for microbial activity, which was determined to be 10% (see Methods). The proportion of days during the typical 240 day wet season in which microbes would be active (WETDAYS) was estimated from the average annual rain fall from the regression in Fig. 3C:

$$WETDAYS = (0.00044 * RAIN - 0.025) * 240 \tag{9}$$

WETDAYS can be multiplied by the maximum rate of microbial respiration (Fig. 3D), as determined by lab incubations of Serengeti soils:

$$MRESP_t = WETDAYS * (0.7 + 0.3 * SAND\%/100) * (0.00044 * SOC_t - 0.579), \tag{10}$$

where the expression $(0.7 + 0.3 * SAND\%/100)$ corrects the respiration rate for greater accessibility to SOC in sandy soils, as suggested by data in *Ruess & Seagle (1994)*.

These equations account for all the major inputs and losses of stable C into soil. The *change* in sequestered carbon ($\Delta SOCt$) can be estimated by adding the plant derived SOC ($PDSOC_t$) to the dung derived SOC ($DDSOC_t$) and then subtracting the carbon lost through microbial maintenance respiration ($MRESP_t$) as follows:

$$\Delta SOCt = PDSOCt + DDSOCt - MRESPt. \tag{11}$$

By setting $\Delta SOCt = 0$, the above equation can be solved for the $SOCt$ term in $MRESPt$ to yield an equilibrium $SOC_{eq}$.

$$\begin{aligned}
SOC_{eq} &= [PDSOCt + DDSOCt + WETDAYS * (0.579) \\
&\quad * (0.7 + 0.3 * SAND\%/100)]/[(0.00044 * WETDAYS \\
&\quad * (0.7 + 0.3 * SAND\%/100)].
\end{aligned} \tag{12}$$

## Model predictions

The equation for $SOC_{eq}$ allows for a relatively straightforward analysis of how different factors influence soil carbon dynamics. It can predict $\Delta SOCt$ from knowledge of some initial soil carbon stock Eq. (11), or it can predict the equilibrium or steady-state carbon stocks following an extended period of certain conditions (grazing, fire, plant species composition, etc.) Eq. (12). My focus here is to explore the influence of different factors on equilibrium carbon stocks, as these represent different states that grassland systems might be approaching from current or past states. This analysis will also reveal the qualitative impacts of different factors on SOC stocks or fluxes.

In the SNAP model, $SOC_{eq}$ increased with higher plant lignin and cellulose content, but decreased with increasing rainfall and soil sand content (Fig. 4). Increasing rainfall decreased C inputs through roots Eq. (6)) and increasing sand reduced water holding capacity and aboveground production as well as increased soil microbial respiration through increased access to organic carbon in coarser soils. In contrast to these

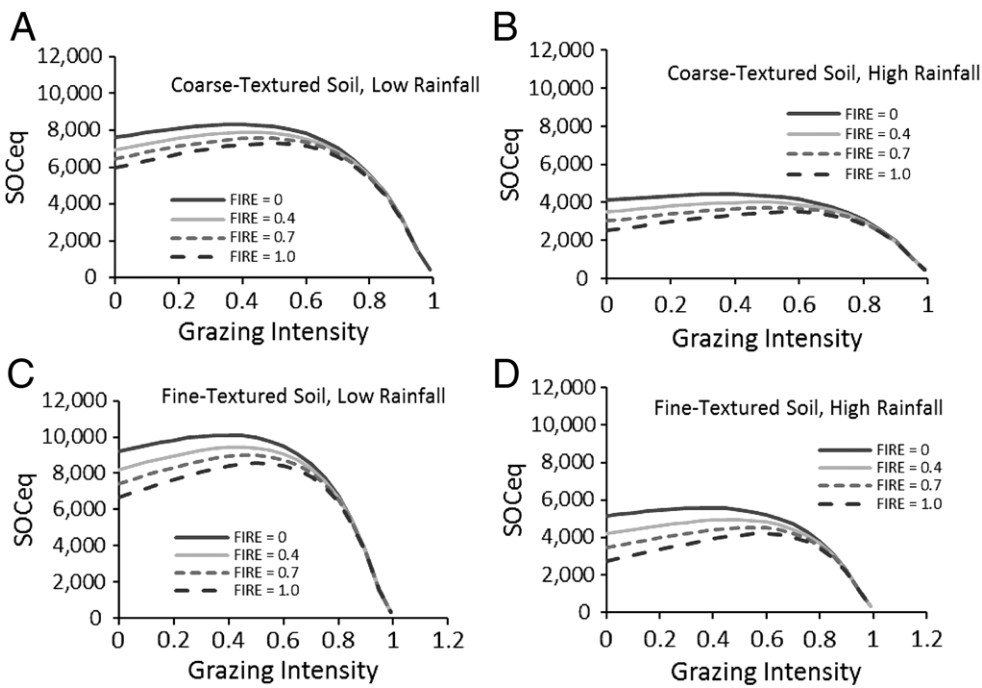

**Figure 4** **Predictions of soil carbon stocks in the Serengeti.** Model predictions of SOCeq stocks to 40 cm depth as a function of the key input variables. These include grazing intensity and fire frequency for different combinations (A–D) of low (RAIN = 450) and high (RAIN = 800) rainfall and low (SAND% = 25, fine-textured soils) and high (SAND% = 65, coarse-textured) soil sand content. Lignin and cellulose (LIGCELL) was set at 0.3 following *Barton et al. (1976).*

straightforward effects, grazing impacts on $SOC_{eq}$ depended on fire frequency (Fig. 4). At low fire frequency, grazing reduced standing biomass and C inputs through plant decomposition, so $SOC_{eq}$ declined very little at lower grazing intensities but exhibited large decreases in $SOC_{eq}$ at grazing intensities >70%. At high fire frequency, high grazing intensities produced the same dramatic drop in SOC, but low grazing intensity led to lower $SOC_{eq}$ as compared to intermediate grazing intensities.

Compensatory maintenance of LAI across a wide range of grazing intensities (Fig. 2B) featured prominently in these predictions because compensatory growth was predicted to maintain C inputs and aboveground biomass fuel for fires even when grazers consume the majority of available biomass. Other work in Serengeti shows that the shift from stems to leaves is chronic as the result from long-term replacements of taller more erect species with shorter, more leafy species (*McNaughton, 1985*; *Anderson et al., 2007*). As several papers outside Serengeti have shown, C assimilation, which is tightly coupled to leaf area, appears to be the major driver of carbon inputs into ecosystems (*Powell et al., 2006*; *Jarlan et al., 2008*; *Cleverly et al., 2013*). Biomass reflects the ratio of C assimilation to turnover, but turnover can reflect allocation from shoots to roots, exudation of C to soil, consumption by herbivores, etc., and biomass is thus not directly proportional to C

**Table 3 Parameter values used in the sensitivity analysis.**

| Level | Rainfall | Fire frequency | Grazing intensity | Lignin and cellulose | % Sand |
|-------|----------|----------------|-------------------|---------------------|--------|
| Low | 450 | 0.25 | 0.25 | 0.15 | 25 |
| Medium | 650 | 0.50 | 0.60 | 0.25 | 45 |
| High | 900 | 0.75 | 0.90 | 0.35 | 70 |

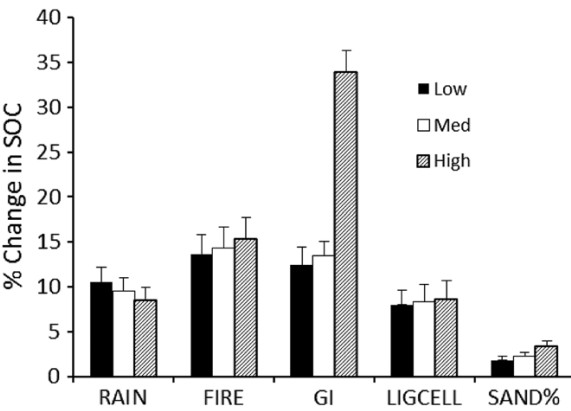

**Figure 5  Sensitivity analysis for the SNAP soil carbon model.** Model sensitivity analysis of the mean (+SE) absolute value of percent change in model-predicted equilibrium SOC (SOCeq). Change results from variation in each input parameter by +10% while holding other parameters constant. Sensitivities (% change in equilibrium SOC ± SE) were calculated for low, medium and high initial values of each parameter (Table 3).

inputs into the ecosystem and especially not the soil. Consequently LAI compensation following herbivory can be reasonably associated with sustained C inputs.

## Sensitivity and uncertainty analysis

The sensitivity of calculated carbon stocks (% change) to 10% changes in parameter values (Table 3) shows that the model predictions are most sensitive to changes in grazing intensity, especially at the upper range of grazing intensities (Fig. 5). $\Delta SOC_t$ was about equally sensitive to changes in rainfall, fire frequency, and lignin and cellulose content, and was least sensitive to changes in soil sand content.

The Monte Carlo analysis yielded consistently small standard errors in estimated $SOC_{eq}$ of around 250–400 g/m$^2$ (around 4%) and an overall uncertainty of around 14% (Fig. 6). These figures are consistent with the sensitivity analysis (Fig. 5) and suggest that errors in SOC predicted by the model are much less than the range of SOC (3000–9000 g/m$^2$) observed across the range of conditions in the Serengeti.

## Model validation

Predicted $SOC_{eq}$ was compared to observed SOC ($SOC_{obs}$) in two ways. First, the average predicted $SOC_{eq}$ and $SOC_{obs}$ were compared at the site level ($N = 8$), i.e., averaged over all three plots at a site, which reflects that RAIN, FIRE, and SAND% were the same for all
three plots. Secondly, to test whether accounting for local variation in grazing intensity and lignin and cellulose (Table 4) affected model success, $SOC_{eq}$ and $SOC_{obs}$ were compared at the plot level ($N = 24$). At the site level, average $SOC_{eq}$ at a site strongly predicted the mean SOC among grazed plots at that site (Fig. 6A): $SOC_{obs} = 0.817$ ($\pm 0.12$ s.e.m.) $SOC_{eq} + 858.1$ ($\pm 1178.3$ s.e.m.). The slope was not significantly different from 1 and the intercept was not different from zero ($t$-tests, $P > 0.5$). Likewise, at the plot level, $SOC_{eq}$ for a plot strongly predicted the mean SOC for that plot (Fig. 6B): $SOC_{obs} = 0.77$ ($\pm 0.11$ s.e.m.) $SOC_{eq} + 1122.1$ ($\pm 1101.3$ s.e.m.). However, even though the intercept was not different from zero the slope was significantly different from 1 and (z-tests, $P > 0.5$). Both slopes were less than 1, which suggests that the model may over-predict SOC under conditions that sustain the highest SOC (drier conditions with finer-textured soils, few fires, and moderate grazing intensities). Very few of the microbial respiration rates measured by *Ruess & Seagle (1994)* occurred under these conditions, and it is possible that respiration rates were higher than those incorporated in the SNAP model, which would result in lower than predicted SOC at equilibrium. Further exploration of plant compensatory responses, soil microbial respiration and other key processes under drier conditions is likely warranted.

The strong correlation with observed carbon stocks (Fig. 6) suggests that the model captures the most important factors influencing soil carbon for the Serengeti system (Fig. 2) and that plant compensatory responses are a key component of soil carbon dynamics in the Serengeti ecosystem. The model is balanced in its sensitivity to different input parameters (Fig. 5), with the exception of its sensitivity to grazing intensity above 70% (Table 3, Figs. 4 and 5). The model has enhanced sensitivity to grazing above this level because LAI declines rapidly under increasing grazing intensity once grazing off-take exceeds plants' ability to trade stem (Fig. 2B) or rhizome (not measured here) to maintain leaf area and hence carbon inputs to the ecosystem.

As is implied by the solution for $SOC_{eq}$ (12) soil carbon stocks are predicted to increase linearly with lower soil sand content, lower rainfall, and greater plant lignin and cellulose. The predicted response of $SOC_{eq}$ to decrease with increasing rainfall appears counterintuitive but results from the decline in root biomass and productivity with increasing rainfall that was observed in the one available tropical grassland study (*McNaughton, Banyikwa & McNaughton, 1998*). This may occur because, within tropical grasslands, at even the lowest rainfall amounts (400–500 mm), aboveground productivity and biomass in the absence of grazing can exceed 600 g/m$^2$ (Table 4) enough to induce light limitation at the soil surface (*Anderson et al., 2007*). Consequently, plants may potentially trade off allocation to root production to compete for light aboveground. This may lead to declining root-derived carbon inputs at higher rainfall (Fig. 3B), and, since roots of herbaceous plants generally contain higher lignin contents than their aboveground tissues, reduced overall stable carbon inputs to soil.

A key and unique aspect of the SNAP model is the interaction between fire and grazing (Fig. 4), which leads to a hypothesized peak in possible carbon stocks at intermediate grazing intensities (Fig. 4) that is amplified by more frequent fires. This is not a dynamic

**Table 4 Measurements and conditions at sites in the 9-year exclosure experiment.** Mean (+ s.e.m., $N = 3$) characteristics of grazed plots at the eight study sites in the grazing experiment.

| Site | Grazing intensity (%) ($N=3$) | Lignin + Cellulose (%) ($N=3$) | Rainfall 1999–2008 (mm/y, $N=9$) | Fires, 2000–2008 | Soil N (%) | Soil C (%) | Soil P (‰) | Sand (%) | Silt (%) | Clay (%) | Bulk Density (g/cm$^2$) |
|---|---|---|---|---|---|---|---|---|---|---|---|
| Balanites | 32 ± 14 | 34.1 ± 2.3 | 721 ± 86 | 4 | 0.19 ± 0.05 | 1.84 ± 0.13 | 32.5 ± 3.0 | 51.0 ± 2.5 | 38.3 ± 4.2 | 10.7 ± 2.2 | 1.31 ± 0.14 |
| Barafu | 65 ± 4 | 34.5 ± 1.9 | 472 ± 31 | 2 | 0.26 ± 0.07 | 3.14 ± 0.06 | 1132 ± 3.8 | 27.6 ± 2.2 | 60.6 ± 4.1 | 11.7 ± 2.0 | 0.85 ± 0.12 |
| Klein's Camp West | 56 ± 9 | 36.3 ± 2.4 | 771 ± 61 | 5 | 0.22 ± 0.03 | 1.77 ± 0.09 | 5.9 ± 0.8 | 40.6 ± 3.1 | 35.5 ± 3.3 | 23.9 ± 2.4 | 1.07 ± 0.17 |
| Kemarische Hills | 66 ± 3 | 31.6 ± 3.1 | 832 ± 87 | 4 | 0.25 ± 0.06 | 2.67 ± 0.08 | 50.0 ± 8.6 | 35.5 ± 2.7 | 52.0 ± 1.8 | 12.5 ± 2.8 | 0.96 ± 0.04 |
| Kuka Hills | 49 ± 4 | 37.7 ± 2.4 | 784 ± 41 | 5 | 0.13 ± 0.04 | 2.13 ± 0.03 | 7.5 ± 0.5 | 45.4 ± 1.2 | 46.2 ± 2.8 | 8.40 ± 1.3 | 1.15 ± 0.12 |
| Musabi Plains | 28 ± 13 | 37.6 ± 1.9 | 885 ± 63 | 7 | 0.14 ± 0.07 | 2.20 ± 0.21 | 69.2 ± 6.3 | 32.9 ± 4.6 | 31.2 ± 3.6 | 35.9 ± 2.9 | 0.90 ± 0.10 |
| Soit Olowotonyi | 54 ± 13 | 35.6 ± 2.1 | 499 ± 39 | 4 | 0.11 ± 0.02 | 1.91 ± 0.14 | 1240 ± 5.2 | 32.1 ± 4.1 | 55.4 ± 4.1 | 12.5 ± 2.4 | 0.84 ± 0.08 |
| Tagora Plains | 69 ± 12 | 31.8 ± 2.8 | 654 ± 87 | 5 | 0.15 ± 0.06 | 1.85 ± 0.07 | 61.2 ± 3.0 | 65.8 ± 3.7 | 28.0 ± 1.5 | 6.21 ± 1.3 | 1.22 ± 0.17 |

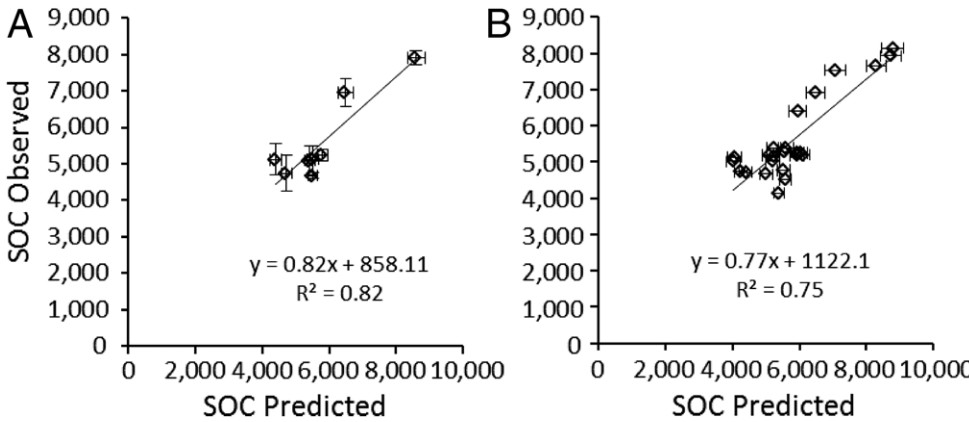

**Figure 6 Validation of the SNAP soil carbon model** Model-predicted equilibrium soil organic carbon stocks (g/m² to 40 cm depth) compared to (A) mean (±s.e.m.) observed soil carbon stocks ($N = 3$) measured at the eight 9-year grazing experiment sites and (B) the 24 grazed plots in the 9-year grazing experiment. Horizontal error bars were derived from Monte Carlo simulations based on parameter errors presented in Table 3.

that is captured explicitly by previous models (*Paustian, Parton & Persson, 1992*; *Parton et al., 1995*; *Pineiro et al., 2010*), and is captured here specifically because of the influence of plant compensation. As expected, at high grazing intensities, the loss of potential fuel diminishes the impact of fire on SOC in the model.

Literature reviews suggest that soil nitrogen (N) dynamics plays a key role in driving grazer effects on soil carbon dynamics (*Derner & Schuman, 2007*; *Pineiro et al., 2010*). Thus, it may be surprising that the model performed well without incorporating any explicit N dynamics. One reason for this is that, across the rainfall gradient in the Serengeti, rainfall and soil moisture are the primary drivers of productivity, with soil N having a much more muted effect (*McNaughton, 1985*; *Holdo et al., 2007*). Second, the adjustment of productivity for soil texture and water holding capacity in the model functions (see Eq. (1)) may have incorporated hidden effects of soil N on carbon dynamics, to the extent that soil N is positively correlated with clay and silt (*Ruess & Seagle, 1994*) and grazing intensity (*Augustine & McNaughton, 2006*). Therefore it seems unlikely that the model fit might be improved by an explicit incorporation of soil N as a variable governing production and microbial abundance and respiration.

Another key factor that is important in most other soil carbon models is temperature, and it perhaps surprising that the model performed well without a response to temperature. In tropical systems like Serengeti, soil temperatures vary from 23°C in the wet season when soils are cooled by evaporation to 26°C in the dry season. This low temperature range may be typical for tropical grasslands (*Caquet et al., 2012*) and is well within the normal active range for many soil microbes. The large variation in rainfall across sites and accompanying very small variation in mean temperature suggests that the principal variable driving soil microbial activity in this system, and perhaps in most

tropical grassland systems, may be soil moisture (*Caquet et al., 2012*; *Manzoni, Schimel & Porporato, 2012*). It is possible that grazing modified soil temperatures more than was assumed here, and feedbacks between grazing, temperature, soil moisture evaporation, and microbial activity might improve the predictability of the model. However, including such dynamics would appear unlikely, from the good fit of the model to data in this paper (Fig. 6) and from previous results (*Wilsey et al., 2002*; *Epron et al., 2004*; *Caquet et al., 2012*), to provide a significant improvement in model predictive power for tropical grasslands.

The mass-balance (SNAP) model developed here contrasts in several ways from current standard models of grazing impacts on SOC, such as CENTURY (*Paustian, Parton & Persson, 1992*; *Schimel et al., 1994*; *Parton et al., 1995*) and the Hurley Pasture model (*Arah et al., 1997*; *Thornley & Cannell, 1997*). First, these previous models are much more complex, as they require many (>20) parameters, such as pH, exchangeable bases, temperature, soil nutrients, plant tissue nutrients, etc., beyond the five parameters required by the SNAP model. The more complex models also consider changes in parameter values over relatively fine time scales (days, weeks, or months), as opposed to the annual pulse-decay cycle of the SNAP model. Moreover, the complex models track two or more SOC pools that differ in their turnover of carbon, while SNAP tracks mainly changes in pools of lignin and cellulose. While CENTURY and the Hurley Pasture models generally predict well changes in temperate grassland SOC related to production and decomposition (*McCulley et al., 2005*; *Pineiro et al., 2010*), they have not been adequately tested for tropical systems and likely contain default functional relationships that do not apply to the tropics. Moreover, their structure clearly does not include plant compensation to grazing, and they are therefore more likely to predict negative impacts of grazing on SOC.

The simplified model and tests presented here suggests that tracking just the stable carbon pool, in this case lignin and cellulose, successfully predicted variation in SOC stocks across the Serengeti landscape (Fig. 6). This is supported by the recent observation (*Frank, Pontes & McFarlane, 2012*) that the vast majority of SOC in grasslands is highly recalcitrant to decomposition, with very slow turnover rates and residence times of 100–1000 years. Lignin and cellulose, the primary stable C forms produced by plants, likely comprise the major input of stable carbon to the soil following decomposition of litter or dung (*Andrioli & Distel, 2008*; *Kirschbaum et al., 2008*; *Semmartin, Garibaldi & Chaneton, 2008*). Dynamics of carbon over short time scales and involving fluxes of labile carbon forms (*Wilsey et al., 2002*) therefore may be less important in determining the long-term dynamics of carbon that dictate carbon stocks. Consequently, the much simpler modeling approach used here might be sufficient to explain variation in carbon stocks.

Work derived from the Serengeti, which is a unique ecosystem with the majority of herbivore biomass in migratory large mammal grazers (wildebeest and zebra), is sometimes viewed as inapplicable to other more common grazing systems, or even other ecosystems in Africa. While the Serengeti certainly does not exactly mimic most livestock-dominated grazing systems, it would be unfair to discount the applicability of

Serengeti data and functional relationships to other ecosystems. First, the Serengeti and its history of grazing research offer the only such data for tropical grasslands for some measures (e.g., root biomass versus rainfall, soil microbial respiration rates versus soil properties, etc.). Secondly, from Table 4, it is apparent that the model was tested across a range of soil types (sandy to silty clay), mean annual rainfall (450–900 mm), and intensity of grazing by both resident non-migratory and migratory grazers. In fact, the site with the highest mean grazing intensity features abundant resident grazers that consistently remove biomass and maintain a grazing lawn (*Anderson et al., 2010*), similar to many livestock systems. Nevertheless, additional tests clearly are needed from resident wildlife grazer systems and free-ranging livestock systems to test the broader applicability of the model and specifically the importance of plant compensation in soil carbon dynamics.

## CONCLUSION

Compensation to grazing by plants in Serengeti National Park occurred across a wide range of grazing intensities, maintaining LAI at the expense of stem biomass up to grazing intensities near 70%. Incorporation of this LAI response into a simplified model of soil carbon dynamics, developed with data from the extensive research history in Serengeti, suggests that plant compensation can allow grazers to increase SOC stocks at intermediate grazing intensities and that this influence is stronger at higher fire frequencies. The model performed very well in predicting current SOC stocks at both a plot and site level. The success of the model indicates the potential utility of tracking the fate of stable carbon (e.g., lignin and cellulose) rather than focusing on fluxes in understanding SOC dynamics.

The predictions of the model, which would likely apply mainly to tropical grasslands, suggest that two factors commonly under management control, grazing and fire, can have large impacts on soil carbon stocks. In particular, there may be potential for land users to (1) reduce fire in lightly grazed, fire-prone areas of Africa and Australia (*Kirschbaum et al., 2008*; *Williams et al., 2008*; *Ryan, Williams & Grace, 2011*; *Woollen, Ryan & Williams, 2012*), and (2) to restore degraded soils in overgrazed areas by better management of livestock rather than de-stocking (*Conant et al., 2001*; *Lal et al., 2007*; *Schipper et al., 2007*). Such practices could help rebuild organic matter and soil carbon across vast areas of the developing world, as tropical grasslands and savannas account for at least 10% of the earth's land surface and occur primarily in developing countries. The simplified model presented here requires only a few measured parameters, so predicting changes in accordance with altered land use may be more feasible than using currently popular models, like CENTURY, with large numbers of parameters. The model presented here (Fig. 1), with its inclusion of plant compensation to grazing, may help explain why grazing in C4 grass-dominated grasslands can help sequester soil carbon (*Derner & Schuman, 2007*; *McSherry & Ritchie, 2013*) and may advance our understanding of soil carbon dynamics in tropical grasslands.

## ACKNOWLEDGEMENTS

I thank Tim Tear and Wenhong Ma for comments on the manuscript and to Emilian Mayemba for assistance in collecting the field data.

### Funding

The study was funded by US National Science Foundation grants DEB 0543398 and DEB 0842230 and by Syracuse University, Chancellor's Leadership Initiative Fund. The funders had no role in study design, data collection and analysis, decision to publish, or preparation of the manuscript.

### Grant Disclosures

The following grant information was disclosed by the authors:
US National Science Foundation grants DEB 0543398 and DEB 0842230.
Syracuse University, Chancellor's Leadership Initiative Fund.

### Field Study Permissions

The following information was supplied relating to ethical approvals (i.e., approving body and any reference numbers):

Tanzania National Parks, Tanzania Commission for Science and Technology and Tanzania Wildlife Research Institute provided necessary permits (COSTECH permit number 2011-52-ER-2004-162) and access to the research site and facilities.

### Competing Interests

The results of this work will support efforts by the author to develop consulting and for-profit carbon projects. Syracuse University has filed a provisional patent application for any commercial use of the model.

### Author Contributions

- Mark E. Ritchie conceived and designed the experiments, performed the experiments, analyzed the data, contributed reagents/materials/analysis tools, wrote the paper.

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
