# Peer review of "Plant compensation to grazing and soil carbon dynamics in a tropical grassland"

_PeerJ, doi:10.7717/peerj.233_

## Round 0.1 · original submission · Major Revisions

Methods need to be explained more clearly, with additional details. Make sure to adequately address all the concerns raised by the two reviewers - particularly those by Reviewer #1.

Reviewer 1 ·

Basic reporting

The figures are not well prepared.
1) Figure captions are not in agreement with the figures. The caption for Figure 1 and 2 should be exchanged. (page 37)
2) For Fig. 3, A, B, C and D are not displayed in the figure as mentioned in the caption. (page 41).
3) For Fig. 4, no legend for four lines both in figure and in caption. But this is critical for assessment the predictions of this model. Also, A, B, C and D are not displayed in the figure as mentioned in the caption. (page 42). ‘Coarse Soil’ should by replaced by ‘Coarse-Textural Soil’ for consistence with other figures.

Experimental design

The description for methods is not sufficient. How do you treat the soil samples before analyzing SOC concentrations? Do you remove the roots from the soil samples? Why you chose a depth of 40 cm soil, why not 1 m as most studies?

Validity of the findings

No comments

Additional comments

This study deals with the effects of grazing on soil organic carbon (SOC) in a tropical grassland ecosystem by developing a model, called SNAP, in which the plant compensatory responses to grazing are incorporated as a parameter. Based on this model, the author suggests that plant compensation to grazing may be an important mechanism for sustaining the concentration of SOC in these ecosystems. This is an important topic and one for which we certainly lack the kind of information the author intends to convey. My major concerns are the development of the model ‘SNAP’ and its validity.
This model is based on a conceptual framework (Figure 1) and some established relationships between variables used in the model (Figure 2 and 3). However, some steps in the process of ‘SNAP’ development need to be further clarified.
First, in equation (2) (page 16, line 11) ANPPtest = RLA*ANPPtmax, relative leaf area index (RLA) are used, but in the combined equation (5)(page 17,line 1), RLA is replaced by leaf area index (LAI)? What is the difference between RLA and LAI? And if RLA is replaced by LAI, is equation (2) still correct? Second, equation (10) (page 18, line 13),
MRESPt = WETDAYS* (0.7+0.3SAND%/100)*(0.00044*SOCt-0.579), is based on Figure 3D, however, the source of (0.7+0.3SAND%/100) is not clear. What does this part mean?
Another key issue is the validity of this model. As mentioned by the author, this model is simpler in relative to other models. Some important factors, such as temperature and nitrogen are not included in this model. The explanation for the lack of these parameters is not strong enough. For soil nitrogen, the author only mentioned that the effect of nitrogen on primary productivity was not significant in this system. However the effects of nitrogen on the structure and function of microbial community should be strong, which may affect microbial respirations profoundly. Furthermore, Based on the relationships between variables involving in this model, the uncertainty of this ‘SNAP’ model is large. The R2 for the relationship between LAI and GI is 0.58 (Figure 2A), suggesting uncertainty for the use of equation (3) (page 16, line 16) is more than 40%. Also, the R2 for the relationship between PL and GI is 0.65 (Figure 2B), suggesting about 35% uncertainty for the use of equation (4) (page 16, line 20). Therefore, the combined equation (5) should have much higher uncertainty.

Minor Points:
1) Page 4, line 8: Replace ‘Leaf area index (leaf area/area sampled), or LAL’ to ‘Leaf area index (LAL, leaf area/area sampled).
2) Page 5 line 11: Remove ‘and’.
3 Page 5 line 22: Insert ‘from’ between ‘gathered’ and ‘a long-term’.
4) Page 7 line 6: Insert ‘(SNP)’ between ‘Serengeti National Park’ and Tanzania.
5) Page 7 line 7: Remove ‘Serengeti National Park (SNP) features.’
6) Page 15 line 13: you mentioned five variables, but you only list 4 variable, what is the rest one?
7) Page 15 line 23: Insert ‘AWHC’ after ‘adjusted water holding capacity’.
8) Page 19 line 10: ‘SOCeq increased with higher plant lignin content’, you do not measure the content of lignin, but the content of lignin and cellulose in total, right?

·

Basic reporting

The manuscript is well written. The structure, figures and tables are all described clearly. But there is some minor mistakes in the Figure legends and articles.

Experimental design

The experimental and collected data are reasonable. Methods have been described clearly. But it is likely more reasonable to include some other key factors in developing the model.

Validity of the findings

The data and the results in the manuscript is basiclly credible and acceptable. The findings in the article is important to understand the impact of hervivores in tropical grasslands and less reported in previous studies. The conclusions is also appropriately stated. But parts of the results and conclusions of this study may need to be careful to weigh and consider. In fact, the simle model compared with other complex models may be easy to predict the SOC dynamics, but is likely neglect some the complex processes and interactions among different factors in carbn flux.

Additional comments

This manuscript studied the plant compensatory response to grazing in the tropical grasslands and conducted a simple model to predict the SOC change. The data base and its analysis as presented in this manuscript are very useful for us to understand how plant response to herbivores and fire in grasslands and predict the SOC dynamics use simple variables. The results seem to be supported by the data. However, it is still not entirely clear due to lack of some more reasonable explanations or assumptions in developing the SNAP models, particularly for the models without including some key factors rather than using some single variables to be the base of SOC dynamic model. Additionally, there are some things I like about this manuscript and some problems with the ms in its current form.
1. Abstract: Compensation is an important responses of plants to grazing, which including many compensatory features, such as mentioned in this article (i.e., sacrificing stems for leaves), over-compensation or equal-compensation in total biomass, etc. The author here described the compensation of plants by maintain leaf area potentially increase SOC in tropical grasslands. However, does the total plant production increase in the intermediate grazing intensity in the tropical grasslands? We may not to be exact to indicate that the compensation by LAI might increase SOC if without over-compensation occur in total plant production. The total biomass may decline due to the decreased stem production although leaves stimulated by intermediate grazing. Thus, the C inputs of SOC stock is actually reduced due to less plant issues.
2. Material and methods:
P8 line 11-13: The tropical grasslands mix by different plant species, which is expected to respond differently to grazing due to the preferences by herbivores. Some species might not be browsed by herbivores. How did LAI was measured? Was it measured by averaging all plants species in each 15 * 15 cm quadrat? The residents in tropical grasslands are usually large herbivores, which have large movement and may lead to patches. As it , is the quadrats by 15*15cm big enough to sample the plant species with compensatory response?

P 8 line 10 Compensatory response to intermediate intensity grazing was measured in a short-term experiment in this article. Is there other research articles or authors previous work have indicated that this compensatory response is a long-term response or the compensatory response will change as the climate change? As the SNAP model developed here is potentially used to predict the SOC dynamics rather than reductive evaluation of the impact of possible plant compensation, it is better to explain if the compensatory response is a general feature. Will SNAP model be used to predict the SOC dynamics if it is just a short-term plant response to grazing?
3. Results and discussion:
P14 line 7-9 The article conducted the regression between LAI and grazing intensity (GI) and used the relationship (Fig. 2) to develop the SNAP model. Does their relationship change with the climate change between years or is there other climatic factors affect LAI besides of GI? For example, when dry years, does leaf shrink its area in spite of grazing intensity?
P15 line 1-9 The key analyses of literature data to build empirical relationships used for model parameters (Fig. 3) were all conducted between simple factors, such as root production- rainfall, respiration-SOC. Did the following models based on these simple regression relationships neglect other impacting factors or interactions between simples factors?
The SNAP model in the article did not include the important temperature variable. However, in many ecosystems, air temperature or soil temperature is an important and key driver to carbon fluxes, especially to plant decomposition, soil respiration, which determine the SOC balance. Although some explanation in the discussion (Page 22, line 12) on the potential effect of temperature on the modeling result, it is better to include this variable (it is likely not hard to collect the temperature information) in the models, at least to conduct some analysis how temperature or soil temperature affect plant production, root and microbial respiration, LAI.
The parameter values in the sensitivity analysis was used to evaluate the model predictions. Is there any rules or standards on the classification of the variables in Table 2, especially GI?
P16 line 4-7: The author in the article used the adjusted McNaughton’s (1985) productivity relationships including soil texture and got the statistical equation formula. Why did such adjustment? Is there relationship between ANPP and rainfall and sand%?
4. Other minor mistakes:
P14 line 13-15: “These two patterns support the hypothesis that plants maintain C inputs to the ……” is there any reference about this hypothesis?
P16 line 15: “…….in this study (Fig. 1A)”: should be Fig. 2A?
P16 line 19: “…….in this study (Fig. 1B)”: should be Fig. 2B?
P16 line 16: why is the formula “LAI = PL/0.6 - 0.031*EXP(4.3*GI)”, but the Fig. 2A indicated “LAI = 1.2 - 0.031*EXP(4.3*GI)”?
P16 line 20: why is the formula “PL =0.6 + 0.35* GI”, but the Fig. 2B indicated “PL =0.6 + 0.24* GI”? Please correct the formula in the Figure 2 or in the article and the following results based on the formulas!
P 20 line 15 and 18: the formulas of SOCobs were not fitted the Fig. 6. In page 44. It seemed reverse to the formulas in the Fig. 6.
P 40 Fig. 2 The legend of the Fig. A and B was reverse.

---

## Round 0.2 · accepted · Accept

Thank you for addressing all the comments by the two reviewers in your revised version. This is a nice piece of work, shedding new light on the effects of grazing on soil carbon dynamics.